# COVID-19 Mortality in the Colorado Center for Personalized Medicine Biobank

**DOI:** 10.3390/ijerph20032368

**Published:** 2023-01-29

**Authors:** Amanda N. Brice, Lauren A. Vanderlinden, Katie M. Marker, David Mayer, Meng Lin, Nicholas Rafaels, Jonathan A. Shortt, Alex Romero, Jan T. Lowery, Christopher R. Gignoux, Randi K. Johnson

**Affiliations:** 1Department of Epidemiology, Colorado School of Public Health, Aurora, CO 80045, USA; 2Human Medical Genetics and Genomics Program, University of Colorado Anschutz Medical Campus, Aurora, CO 80045, USA; 3Department of Biomedical Informatics, University of Colorado Anschutz Medical Campus, Aurora, CO 80045, USA; 4Colorado Center for Personalized Medicine, Aurora, CO 80045, USA

**Keywords:** COVID-19, COVID-19 related mortality, causes of death, cardiovascular disease, diabetes, respiratory disease

## Abstract

Over 6.37 million people have died from COVID-19 worldwide, but factors influencing COVID-19-related mortality remain understudied. We aimed to describe and identify risk factors for COVID-19 mortality in the Colorado Center for Personalized Medicine (CCPM) Biobank using integrated data sources, including Electronic Health Records (EHRs). We calculated cause-specific mortality and case-fatality rates for COVID-19 and common pre-existing health conditions defined by diagnostic phecodes and encounters in EHRs. We performed multivariable logistic regression analyses of the association between each pre-existing condition and COVID-19 mortality. Of the 155,859 Biobank participants enrolled as of July 2022, 20,797 had been diagnosed with COVID-19. Of 5334 Biobank participants who had died, 190 were attributed to COVID-19. The case-fatality rate was 0.91% and the COVID-19 mortality rate was 122 per 100,000 persons. The odds of dying from COVID-19 were significantly increased among older men, and those with 14 of the 61 pre-existing conditions tested, including hypertensive chronic kidney disease (OR: 10.14, 95% CI: 5.48, 19.16) and type 2 diabetes with renal manifestations (OR: 5.59, 95% CI: 3.42, 8.97). Male patients who are older and have pre-existing kidney diseases may be at higher risk for death from COVID-19 and may require special care.

## 1. Introduction

Since its appearance in 2019, COVID-19 has caused 668 million reported infections and 6.73 million deaths worldwide, as of 18 January 2023 [1,2]. These numbers continue to climb, albeit at a slower rate, even with the availability of vaccines and booster shots that became available in 2021. The case fatality rates of COVID-19 vary by country, from 1.2% in the United States to as high as 18.2% in Yemen [2]. Despite concerted efforts to understand and prevent death from COVID-19 by the scientific and medical communities, the lack of integrated data sources limits the ability to evaluate risk factors in a complete fashion. These integrated data sources are needed to investigate multiple aspects of a patient’s health to dive deeper into who is dying from COVID-19, ultimately helping to create more targeted healthcare approaches.

Prior studies have identified several demographic factors such as age, sex, and race-ethnicity that influence COVID-19 mortality. Men are dying at higher rates than women, suggesting sex may be an important factor. According to Global Health 50/50 data, men make up the largest percentage of deaths from COVID-19 in the majority of countries, even in countries predominantly occupied by women [3]. Age has also been identified as a risk factor for COVID-19 mortality. Those under 18 make up a very small percentage of those who have died from COVID-19, with the majority not requiring hospitalizations [4]. However, older patients appear to be at a much higher risk of dying from their infection. A Morbidity and Mortality Weekly Report (MMWR) released in early 2020 showed that 46% of patients who died from COVID-19 were between 65 and 84 years old [5]. Race and ethnicity have also been shown to be related to the rate of mortality from COVID-19. A study performed in New York City during the height of the COVID-19 pandemic identified that Blacks were nearly twice as likely to be hospitalized for COVID-19, while Hispanics were more likely to experience in-hospital mortality from COVID-19, as compared with Non-Hispanic White patients [6].

Other studies have identified pre-existing health conditions as risk factors for COVID-19 mortality, though study findings are inconsistent. According to an early study from the Chinese Center for Disease Control and Prevention, case-fatality rates for COVID-19 were higher in patients with pre-existing health conditions, with the highest rates for cardiovascular disease (10.5%), closely followed by diabetes (7.3%) and COPD (6.3%) [7]. A paper published in April 2022 demonstrated that underlying diabetes was significantly associated with a two-fold increase in COVID-19 mortality [8]; however, a more recent study published in July 2022 showed no significant difference in COVID-19 mortality among intensive care patients with diabetes and those without [9]. In addition to diabetes, pre-existing cardiovascular conditions have been linked to patients dying from COVID-19. A systematic review of articles published between January and April 2020 demonstrated a significant positive association between the presence of cardiovascular comorbidities that included hypertension and coronary heart disease [10]. More recently, hypertension has been shown to be associated with death from COVID-19 in the elderly population [11]. COPD was also found to be significantly associated with COVID-19 mortality in a meta-analysis in January 2020 [12].

In addition to pre-existing conditions, previous research has investigated the possible link between COVID-19 and the gene that determines blood type, ABO, but the study results vary. Several studies have suggested there is a correlation between blood type and risk of transmission or severity [12,13]. However, other studies have conversely shown that blood type has no effect on mortality [13,14].

While vaccination adoption has helped to reduce mortality rates, the continued presence and changing nature of COVID-19 make it crucial to understand who is most affected by COVID-19 mortality and what other factors could affect the risk of death. This study expands knowledge on the determinants of COVID-19 mortality in Colorado using a unique population that integrates genetics with electronic health records (EHR) data. Using the Colorado Center for Personalized Medicine (CCPM) Biobank as an integrated data source, we were able to comprehensively evaluate pre-existing risk factors for COVID-19 mortality. In addition to characterizing case fatality and mortality rates for COVID-19, pre-existing health conditions based on phecodes as well as blood type were analyzed as risk factors for COVID-19 mortality while controlling for sociodemographic factors.

## 2. Materials and Methods

### 2.1. Study Population

We conducted a retrospective cohort study of participants of the CCPM Biobank. The full cohort selection is shown in Figure 1. The Biobank at CCPM is a collaboration between the University of Colorado Health (UCHealth) and the University of Colorado Anschutz Medical Center and School of Medicine, as described in detail elsewhere [15]. Study consent for the CCPM Biobank is accessible on My Health Connection, the patient portal for UCHealth. All UCHealth patients who are 18 years of age or older and can provide consent for themselves are able to opt-in to the program. CCPM Biobank participants live in all 50 states as well as the District of Columbia, although the majority (~67%) reside along the Front Range region of Colorado, which includes the Denver Metropolitan Area, Fort Collins, and Colorado Springs. Participants are predominantly women, non-Hispanic White, and mostly uniformly distributed for age groups 30 to 69, while somewhat lower for 18–29 and 70+. The full participant demographic breakdown was made available by Johnson et al., 2022 [16].

De-identified data for studies of COVID-19 were provided by the Health Data Compass data warehouse, using unique masked IDs to connect electronic health records (EHR) information with genetic data and mortality for CCPM Biobank participants. This study population included only Biobank participants (enrolled as of 27 July 2022) who were active in the UCHealth system prior to COVID-19—where active was defined as having at least three face-to-face encounters with the UCHealth system—in any capacity which could range from routine physicals to chronic care, between January 2015 and January 2020.

### 2.2. Pre-Existing Conditions

Phecodes, a high-throughput phenotyping tool to define disease status and our main exposure of interest, were constructed from ICD-9 and ICD-10 billing codes using the PheWAS R package [17]. A person was considered to have been diagnosed with a pre-existing condition if they had at least three phecode occurrences in their EHR within a COVID-19 pre-existing time frame window (January 2015 to January 2020), similar to the methodology of Salvatore et al., 2021 [18]. Those that had one or two phecode occurrences within the pre-existing time frame window were removed from the analysis for that specific phecode and COVID-19 diagnosis association. This was carried out to exclude possible misdiagnoses or technical errors.

The main explanatory variables investigated were pre-existing cardiovascular diseases, diabetes, and respiratory diseases. These were selected a priori based on prior literature. The specific phecodes from these three categories were selected for analysis if they had at least 10 deaths from COVID-19. In addition, 12 phecodes that had at least 10 deaths from COVID-19 and were not previously reported in the COVID-19 literature were examined as “control” phecodes. These control phecodes were included to establish a benchmark in which to compare the outcomes and would support the validity of any correlations between the explanatory phecodes and COVID-19 mortality. Using these unrelated pre-existing conditions ensures there is not a selection bias for pre-existing conditions as a whole. A list of the 289 phecodes compiled (both those selected for analysis and those not selected) can be found in the Appendix A and includes 172 cardiovascular disease phecodes, 20 diabetes phecodes, 85 respiratory disease phecodes, and 12 control phecodes.

### 2.3. Genotyping

A subset of the CCPM Biobank has genetic data currently available as described in detail in Wiley et al. Briefly, genotyping was performed for 34,435 Biobank samples on the Illumina MEGA-EX and MEGA-Plus platforms. Sample QC was performed by plate checking for sex discordance, excess heterozygosity, high missingness, genetic duplicates, and control sample concordance. SNP QC checking was performed for high missingness, high differential in allele frequency by sex, high differential in allele frequency by plate, and high discordance across 5 control samples consistent across all plates. Strand ambiguous, monomorphic, and SNPs with ancestry-specific Hardy–Weinberg equilibrium *p* < 1 × 10^−12^ were removed prior to imputation using the TOPMed imputation server [19,20]. Genotyped SNPs and imputed SNPs with Rsq > 0.7 were retained for use in analyses.

Genetically predicted ABO blood types were derived from all Biobank participants with genotype data based on haplotypes of three SNPs, following the methodology of McLachlan et al. [21]. The combination of genotypes of rs8176747, rs8176746, and rs8176719 were used to determine ABO on each haplotype, defined as (in the SNP order listed previously): GCG -> “A”, GAC -> “B”, and deletion at rs8176719 -> “O”. Due to the lower number of COVID-19 deaths among those with known blood type, only predicted blood groups A (e.g., AA, AO) and O were reported and analyzed. Due to low frequency, a Fisher’s exact test was used to test the significance of the four blood type groups.

### 2.4. Mortality and Other Covariates

The Health Data Compass clinical data warehouse captures the mortality of UCHealth patients through routine linkage with vital records (death certificates) data provided by the Colorado Department of Public Health and Environment (CDPHE). Available mortality information included cause of death as certified by a physician or coroner/medical examiner, corresponding ICD-10 codes generated by the Centers for Disease Control and Prevention, and age at death. The response variable, death from COVID-19, was classified as a patient either having the ICD-10 code for COVID-19 (U07.1) coded as a cause of death or having “COVID-19” or its synonymous names (SARS-CoV-2, nCOV, etc.) as contributing factors of death. COVID-19 infection was defined as those with a diagnostic ICD-10 code U07.1 in the EHR or who died from COVID-19. Due to ~3 month lag in death certificate registration, cause of death coding, and clinical data updates, mortality is expected to be ~95% complete. The study period included COVID-19 cases and deaths between January 2020 and July 2022.

The covariates included as possible confounders included age, sex, healthcare utilization (HCU), and race/ethnicity. Given the de-identified nature of the dataset, all analytical variables were categorized as needed to maintain >10 subjects in each level to protect patient privacy. Age, defined by years since birth at the July 2022 data freeze, was taken from the EHR and categorized as follows: 18–49, 50–64, 65–79, and 80+. Sex (male or female) was also obtained from the EHR. HCU was defined by the number of face-to-face visits between January 2015 and January 2020, with three being the minimum number based on selection criteria. HCU was modeled in quintiles due to the strong right skew of the distribution, with categories (and cut points) at HCU-1 (3–10 visits), HCU-2 (11–22 visits), HCU-3 (23–42 visits), HCU-4 (43–85 visits), and HCU-5 (>85 visits). For the pre-existing health condition analyses, race and ethnicity were taken from the EHR and combined into a single race/ethnicity variable (Non-Hispanic White, Non-Hispanic Black, Hispanic, or Other) due to the low-frequency sample sizes among racial-ethnic minority Biobank participants. For the blood type analyses that were conducted on participants with genetic information, ancestry principal components (PCs) were used as a better approximation for race/ethnicity than what was self-reported in the EHR. We were unable to include education and income level in statistical analyses as they were only available at aggregate levels based on a patient’s 3-digit zip code of residence.

### 2.5. Statistical Analyses

All data analyses and figures were produced using SAS 9.4, R Studio, and R version 4.2.0. Case fatality rates for COVID-19 were calculated as the deaths due to COVID-19 in the Biobank divided by the total number of people in the Biobank who were diagnosed with COVID-19. Case fatality for COVID-19 was also calculated among those with cardiovascular disease, diabetes, and respiratory disease as sub-groups. The period mortality rate for COVID-19 was calculated as the deaths due to COVID-19 in the Biobank during the study period divided by the total Biobank population. These values were multiplied by 100,000 to obtain the number of dead per 100,000 people. This was repeated to calculate the mortality rates for cardiovascular disease, diabetes, and respiratory disease. Descriptive statistics (frequency, mean, SD, %) were calculated to characterize explanatory variables overall and by outcome group. Significance in outcome (death from COVID-19 versus survived COVID-19) was tested using chi-square and *t*-tests for categorical and continuous variables, respectively.

Unadjusted multiple logistic regressions were performed to identify associations between each of the pre-existing health conditions and the risk of death from COVID-19. These models were then adjusted for age, sex, HCU, and race/ethnicity. For these associations, odds ratios and 95% confidence intervals were obtained. Odds ratios were used as a measure of risk for death from COVID-19. Phecodes were tested in aggregate categories (cardiovascular, respiratory, and diabetes) and in smaller classifications for the 61 phecodes that had at least 10 deaths from COVID-19. This included 39 cardiovascular disease phecodes, 7 diabetes phecodes, and 15 respiratory disease phecodes. Appendix A contains all the phecodes as well as highlighting these 61. Due to the deidentified nature of the data, only the year of birth and death were available. This precluded the use of survival analyses as the follow-up period for death from COVID-19 was only two years (2020–2022). Significance was assessed with a Bonferroni correction for the number of phecodes tested in multivariable analyses (*n* = 61) at an alpha of 0.05 (*p* < 0.0008). This was repeated for the control phecode analyses (*n* = 12) at an alpha of 0.05 (*p* < 0.004).

## 3. Results

### 3.1. Descriptive Statistics

A total of 155,859 participants met the inclusion criteria, and 31,482 had genetic data available. Descriptive statistics for the explanatory variables and covariates are shown in Table 1 for the full Biobank population, COVID-19 cases, and by COVID-19 outcome (death or survival). Of the 20,797 participants who were diagnosed with COVID-19, there were 190 deaths from COVID-19. The distributions of age, sex, pre-existing cardiovascular disease, diabetes, respiratory disease, and blood type significantly differed (*p* < 0.05) by COVID-19 outcome. Compared with those who survived COVID-19, those who died from COVID-19 were older (mean of 69.6 years versus 50.8 years, *p* < 0.001), enriched for a higher proportion of male participants (59.5% versus 40.5%, *p* < 0.001), participants with pre-existing cardiovascular disease (80.0% versus 47.6%, *p* < 0.001), pre-existing diabetes (38.4% versus 16.7%, *p* < 0.001), and pre-existing respiratory disease (62.1% versus 45.7%, *p* < 0.001). Due to the low number of participants with genotyped B and AB blood types, the number of COVID-19 cases and deaths for these groups were not able to be reported in order to maintain anonymity, however, the Fisher’s exact test suggested significance (*p* < 0.001).

### 3.2. Mortality and Case Fatality Rates

The mortality rates calculated are shown in Table 2. The all-cause mortality rate for the Biobank was 3442 per 100,000 persons (5334 deaths out of 155,859 at risk). The cause-specific mortality rate from COVID-19 in the Biobank was 122 deaths per 100,000 (190 deaths from COVID-19 out of 155,859 at risk). The cause-specific mortality rate from cardiovascular disease was 318 deaths per 100,000 persons (495 deaths from cardiovascular disease out of 155,859 at risk). The cause-specific mortality rate from diabetes in the Biobank was 68 deaths per 100,000 persons (106 deaths from diabetes out of 155,859 at risk). The cause-specific mortality rate from respiratory disease in the Biobank was 138 deaths per 100,000 persons (215 deaths from respiratory disease out of 155,859 at risk).

The results for the case fatality rates are shown in Table 3. The case fatality rate for COVID-19 in the entire Biobank population was only 0.91% (190 deaths from COVID-19 out of 20,797 with COVID-19). Case fatality was much higher among participants with pre-existing chronic conditions. The case fatality rate for COVID-19 was highest among those with diabetes at 2.08% (73 deaths from COVID-19 out of 3508 with COVID-19 and pre-existing diabetes), followed by cardiovascular disease at 1.53% (152 deaths from COVID-19 out of 9951 with COVID-19 and pre-existing cardiovascular disease) and respiratory disease at 1.24% (118 deaths from COVID-19 out of 9532 with COVID-19 and pre-existing respiratory disease).

### 3.3. Multiple Logistic Regression

The results of the adjusted logistic regression models for the 61 individual phecodes with at least 10 COVID-19 deaths are shown in Figure 2. Of the 39 cardiovascular phecodes tested, seven were found to be significant after Bonferroni correction, at *p* < 0.0008. The phecodes with the largest odds ratios were hypertensive heart and/or renal disease (OR: 8.45, 95% CI: 4.83, 15.13) and hypertensive chronic kidney disorder (OR: 10.14, 95% CI: 5.48, 19.16). Of the seven diabetes phecodes, three were positively associated with COVID-19 mortality. The diabetes phecode with the largest increase in risk for death from COVID-19 was type 2 diabetes with renal manifestations (OR: 5.59, 95% CI: 3.42, 8.97). Of the 15 respiratory disease phecodes tested, four were positively associated with COVID-19 mortality. The respiratory phecodes with the largest effect sizes were dependence on a respirator, ventilator, or supplemental oxygen (OR: 6.68, 95% CI: 3.49, 12.24) and respiratory failure (OR: 6.67, 95% CI: 4.24, 10.36).

None of the 12 control phecodes were found to be significantly associated with COVID-19 outcomes after the Bonferroni correction, at *p* < 0.004 (Appendix A). When examined in aggregate, any pre-existing cardiovascular disease (OR = 1.35, 95% CI: 0.9, 2.03) and any pre-existing respiratory disease (OR = 1.09, 95% CI: 0.79, 1.51) were not significantly associated with increased risk of COVID-19 death, while any pre-existing diabetes (OR = 1.40, 95% CI: 1.03, 1.92) was found to be significantly associated by a very small degree (Appendix A).

## 4. Discussion

Among participants of the CCPM Biobank who were diagnosed with COVID-19 in the UCHealth system, 190 (0.91%) died from COVID-19. This case fatality rate was higher among those with pre-existing cardiovascular disease (1.55%), diabetes (2.13%), and respiratory disease (1.25%). Those who died from COVID-19 were more likely to be older and men compared with those who survived COVID-19. After adjustment for demographic factors, we identified 14 pre-existing conditions (phecodes) significantly associated with death from COVID-19, including several related to renal and kidney dysfunction.

Our findings support what has been observed in other COVID-19 studies. Sex appears to be a relevant factor for COVID-19 outcomes, as seen by the higher proportion of men who died, despite making up a minority of the study population. The difference in outcomes based on sex has been apparent since the beginning of the pandemic. Previous studies suggest that this difference could be due to a combination of biological and social factors. Cardiovascular disease predominately occurs in men, specifically in those over 50 years, a comorbidity already linked to COVID-19 mortality [7,10,11]. In addition, outside of healthcare support jobs, men make up a large percentage of the labor force in positions where the COVID-19 mortality risk is higher due to the inability to work from home [22]. For example, men account for 96% of construction and repair work and 75.6% of agricultural work in the United States [23]. This information can be used to further direct COVID-19 prevention, specifically toward men who are at a higher risk for exposure and poor health outcomes.

The COVID-19 mortality rate of 122 deaths per 100,000 persons in the study population was greater than the Colorado COVID-19 mortality rate as of April 2022, which was 88 deaths per 100,000 [24]. The mortality rates per 100,000 for cardiovascular disease, diabetes, and respiratory disease in the Biobank (318, 68, and 138, respectively) were all greater than the mortality rates for Colorado as of 2020 (137, 20, and 42, respectively) [25]. These data may suggest that the Biobank population is sicker than the state population, likely resulting from the fact that Biobank participants are UCHealth patients and must be using the UCHealth system frequently to meet inclusion criteria. Additionally, since COVID-19 cases have been undertested and underreported throughout the pandemic, coupled with the fact that many people did not receive their COVID-19 tests through UCHealth, it is likely that our case count is a substantial underestimate. The methods used for testing for COVID-19, which was not data available to us, could also have affected the number of cases, depending on the accuracy of the tests. We had near-complete ascertainment of death and cause of death, as this endpoint is generated through linkage with vital statistics at the CDPHE. Therefore, our case fatality is probably an upper limit as compared with the overall reported rates for Colorado. However, it should be noted that there was a shorter follow-up time for COVID-19 cases occurring near the end of the study period, and therefore our data set could be missing COVID-19 deaths that occurred right after our cutoff date or COVID-19 deaths that were incorrectly or incompletely captured in vital statistics.

The percentage of those who died from COVID-19 and also had pre-existing cardiovascular disease, diabetes, and respiratory disease (80.0%, 38.4%, and 62.1%, respectively) were double to triple the percentage of those in the Biobank with these conditions as a whole (40.8%, 13.4%, and 32.2%, respectively). This supported previous research that found that these preexisting conditions were more prevalent among those who died from COVID-19 [7]. This is further explained by the results of the multivariable regression models.

There was not a significant association between aggregate categories of cardiovascular disease and respiratory disease and COVID-19 outcomes, in contrast to research at the start of the pandemic. It is possible that the level of significant risk for COVID-19 mortality among those with one of the two pre-existing conditions has changed from the beginning of 2020 to 2022 due to the introduction of COVID-19 vaccines. Recent studies have shown that despite the emergence of new variants, those who received the COVID-19 vaccine have a decreased risk of death as compared with those who did not receive the vaccine [26]. Therefore, vaccination status may have acted as a precision variable that explains variability in COVID-19 mortality, particularly for cases occurring in the latter half of our study period (2020–2022). A weakness of our study is that due to the introduction of COVID-19 vaccines in the middle of our study period, we were unable to investigate the protective effects of the vaccine on COVID-19 mortality or adjust for vaccination status in our analyses. As a result, our associations are likely less precise than if we were able to adjust for vaccination status and may have prevented us from identifying additional associations between pre-existing conditions and COVID-19 mortality. However, given that our exposures of interest (pre-existing conditions) occurred prior to the pandemic and widespread COVID-19 immunization, vaccine status would not operate as a confounder of these associations.

The high percentage of individual phecodes that were significantly associated with a negative outcome reflects previous research that looked more closely at specific diseases such as coronary heart disease [24], rather than cardiovascular disease as a whole. Due to the wide range in phecodes for each category of disease, it is possible that the effects of more harmful diseases could be negated by those less harmful when aggregated in the analyses. The phecodes that demonstrated the largest risk for COVID-19 mortality in the cardiovascular and diabetes categories were connected to the kidneys to some degree (hypertensive heart and/or renal disease, hypertensive chronic kidney disease, and Type 2 diabetes with renal manifestations). This aligns with prior research demonstrating that kidney disease is associated with an increased risk of COVID-19 mortality [27,28,29]. A study by Perico et al. describes how COVID-19 creates endothelial damage, complement-associated blood clotting, and systemic microangiopathy which can lead to kidney damage and eventually a cascade of organ failure [30]. Due to the connection between mortality and kidney disease, it is important for future treatment that patients are screened for pre-existing kidney disease as an additional risk factor. There was also a large risk for COVID-19 mortality among several respiratory phecodes (respiratory failure and dependence on respiratory, ventilator, or supplemental oxygen), which supports previous research [12].

The retrospective design is a strength of the study, as the pre-existing conditions occurred in EHR prior to the appearance of COVID-19. For each patient, extra care was used to confirm that they had the pre-existing health condition listed on their EHR, requiring at least three separate records. By limiting these records to the five years preceding the COVID-19 pandemic, it confirmed that none of these conditions were COVID-19 related. By only including those who have had three face-to-face visits with the UCHealth system, we ensured that these were patients that used the UCHealth system as their main medical provider. A limitation of this project was that due to a small number of COVID-19 cases among those with genetic information, more informative analyses were not able to be performed for blood type, and no firm conclusions could be made. However, the use of genetic information in addition to health records may inform future studies. These procedures should be repeated as more genetic data is generated in the Biobank or larger studies with available genotyping and quality mortality data.

## 5. Conclusions

This project adds to the continuously growing pool of information on the epidemiology of COVID-19 mortality using integrated data sources (EHR, vital statistics, genetics) to provide a more complete picture. Patients with pre-existing cardiovascular diseases and diabetes, especially those involving renal complications, and prior severe respiratory dysfunction who present to health systems with COVID-19 might be at higher risk for mortality and should be closely monitored.

## Figures and Tables

**Figure 1 ijerph-20-02368-f001:**
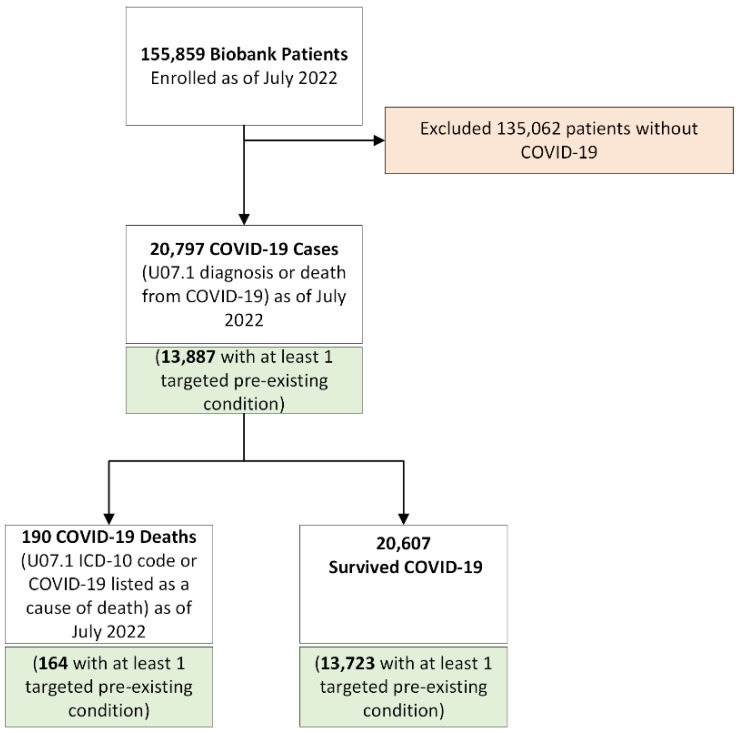
Cohort Selection. Cohort was determined from face-to-face visits and phecode diagnoses.

**Figure 2 ijerph-20-02368-f002:**
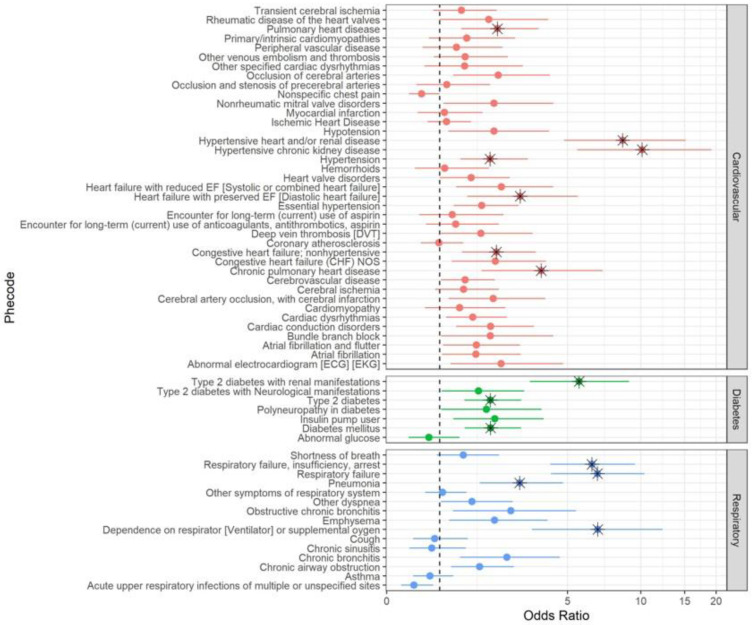
Select Phecodes. Adjusted odds ratios and 95% confidence intervals for risk from death from COVID-19 for selected phecodes. Phecodes with significant odds ratios after Bonferroni correction (*p* < 0.0008) are labeled with an asterisk.

**Table 1 ijerph-20-02368-t001:** Descriptive statistics. This table includes biobank participant characteristics.

	Death from COVID-19 (*N* = 190)	Survived COVID-19 (*N* = 20,607)	COVID-19 Cases (*N* = 20,797)	*p*-Value *	Biobank (*N* = 155,859)
**Age, mean (SD)**	69.6 (12.9)	50.8 (16.0)	51.0	<0.001	52.6 (16.7)
**Age (years), *N* (%)**				<0.001	
18–49	16 (8.42)	10,181 (49.4)	10,197		70,564 (45.3)
50–64	42 (22.1)	5562 (27.0)	5604		40,275 (25.8)
65–79	83 (43.7)	4167 (20.2)	4250		37,710 (24.2)
80+	49 (25.8)	697 (3.4)	746		7310 (4.7)
**Race/ethnicity, *N* (%)**				0.07	
Non-Hispanic White	138 (72.6)	16,025 (77.8)	16,163		125,670 (80.6)
Non-Hispanic Black	14 (7.4)	869 (4.2)	883		6359 (4.1)
Hispanic	30 (15.8)	2597 (12.6)	2627		14,065 (9.0)
Other	8 (4.2)	1116 (5.4)	1124		9765 (6.3)
**Sex, *N* (%)**				<0.001	
Male	113 (59.5)	7282 (35.3)	7395		61,348 (39.4)
Female	77 (40.5)	13,325 (64.7)	13,402		94,511 (60.6)
**Pre-existing conditions, *N* (%)**					
Cardiovascular Disease	152 (80.0)	9799 (47.6)	9951	<0.001	63,518 (40.8)
Diabetes	73 (38.4)	3435 (16.7)	3508	<0.001	20,862 (13.4)
Respiratory disease	118 (62.1)	9414 (45.7)	9532	<0.001	50,230 (32.2)
**Blood type, *N* (%)**				<0.001 **	
A	32 (16.8)	1933 (9.4)	1965		13,378 (8.6)
O	26 (13.7)	1838 (8.9)	1864		13,262 (8.5)
AB	<10	--	--		1363 (0.9)
B	<10	--	--		3479 (2.2)
Not genotyped	125 (65.8)	16,177 (78.5)	16,302		124,377 (79.8)

* Comparing COVID-19 deaths vs. survival. ** Fisher’s exact test, excluding not genotyped.

**Table 2 ijerph-20-02368-t002:** Mortality rates. Mortality rates among 155,859 Biobank participants through January 2022.

	Rate per 100,000, (*N*)
	Biobank
**All-cause mortality**	**3442** (5334)
**Cause-specific mortality**	
COVID-19	**122** (190)
Cardiovascular	**318** (495)
Diabetes	**68** (106)
Respiratory	**138** (215)
Other *	**2777** (4328)

* Mortality rate of all other causes excluding COVID-19, cardiovascular disease, diabetes, and respiratory disease.

**Table 3 ijerph-20-02368-t003:** COVID-19 case fatality. Case fatality overall and by pre-existing condition.

	Had COVID-19	Died from COVID-19	Case Fatality (%)
Biobank population	20,797	190	0.91
**Pre-existing condition**			
Cardiovascular	9799	152	1.55
Diabetes	3435	73	2.13
Respiratory	9414	118	1.25

## Data Availability

The data used for this study are not publicly available to protect the privacy of CCPM Biobank participants. Interested parties may request access to the data through Health Data Compass (https://www.healthdatacompass.org/).

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
