# Peer review of "COVID-19 Mortality in the Colorado Center for Personalized Medicine Biobank"

_ijerph, 2023, doi:10.3390/ijerph20032368_

Round 1

Reviewer 1 Report

The high number of data points, well-defined selection of participants, and well-described data integration approach improve the quality of the manuscript. The findings are helpful to supplement our understanding of the relationship between pre-cardiovascular diseases, respiratory diseases, and diabetes.

Line 62: "A meta-analysis performed in April 2020 concluded that underlying diabetes was significantly associated with a two-fold increase in COVID-19 mortality [8]" Authors of reference [8], mention that "We searched PubMed for case-control studies in English, published between Jan 1 and Apr 22, 2020"; suggesting that the literature search step of the meta-analysis in reference [8] was not performed in April, rather, it included studies published between January and April 2022. There are more recent meta-analyses focussing on the association between Diabetes and COVID-19 fatal outcomes (https://onlinelibrary.wiley.com/doi/10.1002/edm2.338 ; https://pubmed.ncbi.nlm.nih.gov/34697120/, etc. 

Line 167:  "The covariates included as possible confounders included age, sex, healthcare utilization (HCU), and race/ethnicity." : The sentence makes sense, however for more clarity, please avoid repetition of the word "included".

section "2.5 Statistical analyses": Authors described how the case fatality rates were computed and how the logistic regression was performed. In line 150, the authors describe a statistical test used to test the significance of blood group SNP. However, the authors do not describe the statistical test for average age comparison shown in the third line of Table 1.

Table 1: Authors show the distribution of participants' blood types associated with COVID-19 death cases while representing the low number of cases by "<10". Despite the low number of the observed case,  it would be preferable to display the exact number of patients with blood type AB or B for reproducibility purposes. For instance, I was unable to verify fisher's exact test results, because the number of cases with blood type AB and B was missing.

Author Response

Point 1: English very difficult to understand/incomprehensible.

Response 1: The manuscript was reviewed, edited, and approved by the co-authors, all of whom are native English speakers.

Point 2: The high number of data points, well-defined selection of participants, and well-described data integration approach improve the quality of the manuscript. The findings are helpful to supplement our understanding of the relationship between pre-cardiovascular diseases, respiratory diseases, and diabetes.

Response 2: Thank you for recognizing our work’s importance in understanding the relationships between pre-existing diseases and COVID-19 mortality.

Point 3: Line 62: "A meta-analysis performed in April 2020 concluded that underlying diabetes was significantly associated with a two-fold increase in COVID-19 mortality [8]" Authors of reference [8], mention that "We searched PubMed for case-control studies in English, published between Jan 1 and Apr 22, 2020"; suggesting that the literature search step of the meta-analysis in reference [8] was not performed in April, rather, it included studies published between January and April 2022. There are more recent meta-analyses focussing on the association between Diabetes and COVID-19 fatal outcomes (https://onlinelibrary.wiley.com/doi/10.1002/edm2.338 ; https://pubmed.ncbi.nlm.nih.gov/34697120/, etc.

Response 3: Thank you for pointing out the newer citations key to our background. We have updated our citation (line 63) to include the more recent work demonstrating the relationship between diabetes and COVID-19 mortality.

Point 4: Line 167:  “The covariates included as possible confounders included age, sex, healthcare utilization (HCU), and race/ethnicity.” : The sentence makes sense, however for more clarity, please avoid repetition of the word “included”.

Response 4: Edited as suggested, see line 174.

Point 5: section "2.5 Statistical analyses": Authors described how the case fatality rates were computed and how the logistic regression was performed. In line 150, the authors describe a statistical test used to test the significance of blood group SNP. However, the authors do not describe the statistical test for average age comparison shown in the third line of Table 1.

Response 5: We regret the inadvertent omission. We used chi-square (for categorical variables) and t-tests (for continuous variables) to test for significant differences between groups and added this information to the Methods starting at line 205.

Point 6: Table 1: Authors show the distribution of participants' blood types associated with COVID-19 death cases while representing the low number of cases by "<10". Despite the low number of the observed case, it would be preferable to display the exact number of patients with blood type AB or B for reproducibility purposes. For instance, I was unable to verify fisher's exact test results, because the number of cases with blood type AB and B was missing.

Response 6: Cell counts with less than ten patients are censored to preserve patient privacy in accordance with CCPM Biobank policy. Therefore, the exact number of patients for blood types AB and B are not able to be reported exactly and instead must be designated as “<10”.

Reviewer 2 Report

This is a study about covid 19 carried out by the University of Colorado Health and the University of Colorado Anschutz Medical Center and School of Medicine in patients enrolled as of July 2022. Of a total of 158,859 patients, 20,797 were diagnosed with covid 19, of which 190 died from this disease. A descriptive statistical table of the patients is presented, organized paying attention to sex, age, race/ethnicity, pre-existing conditions and blood type, and a detailed analysis of the results obtained is made. In my opinion, this is a work worthy of being published in this journal because it provides important information about this epidemic.

Author Response

Point 1: This is a study about covid 19 carried out by the University of Colorado Health and the University of Colorado Anschutz Medical Center and School of Medicine in patients enrolled as of July 2022. Of a total of 158,859 patients, 20,797 were diagnosed with covid 19, of which 190 died from this disease. A descriptive statistical table of the patients is presented, organized paying attention to sex, age, race/ethnicity, pre-existing conditions and blood type, and a detailed analysis of the results obtained is made. In my opinion, this is a work worthy of being published in this journal because it provides important information about this epidemic.

Response 1: Thank you for the recognition of the importance of this work and its timely contributions to building knowledge around the effects of the COVID-19 pandemic. We appreciate the time you have given to review our manuscript.

Reviewer 3 Report

This study aims to identify risk factors for COVID-19 mortality in the Colorado Center for Personalized Medicine (CCPM) Biobank using integrated data sources, including Electronic Health Records (EHRs).

The great bias of this study is related to the analyzed period: in the tested group, the authors missed indicating the vaccination status. This is an important bias that influences the conclusion of the study. It should be better discussed in the discussion section.

Another important bias is related to the definition of the cause of death: the gold standard method is the autopsy tool (see DOI: 10.3390/medicina57040309). The authors missed describing if they consulted the autopsy report for each case classified as death from COVID. Please, clarify this important aspect. If an autopsy is not performed, discussed this bias in the discussion section.

Finally, the authors missed indicating how the patients were tested to determine the COVID-19 infection: each method could have a bias. If this data is not available, the authors should discuss this important point.

Other points

Introduction section

- Please, update the data reported in line 32 to the end of 2022.

- Please, insert several considerations about the vaccination program: fortunately, it has influenced the mortality rate.

Material and methods:

- Please, insert the start and stop points in the collected data.

Results:

- Table 1, Blood type: <10 -> it is incorrect. Please, insert the exact number.

- Blood type: several important data are missed to achieve a solid scientific conclusion. I suggest removing this section. Alternatively, it should be completely discussed this aspect.

Discussion

- Please, improve this section. Particularly, I suggest inserting a specific section named "Limitations".

Author Response

Point 1: The great bias of this study is related to the analyzed period: in the tested group, the authors missed indicating the vaccination status. This is an important bias that influences the conclusion of the study. It should be better discussed in the discussion section.

Response 1: As pointed out by the reviewer and in our introduction and discussion, the availability of COVID-19 vaccines, which began during our study period (2020-2022), has positively impacted COVID-19 mortality outcomes (lines 33-35, 78-81, 343-348, citation n. 27). Our inability to include vaccination information as a precision variable in analyses is a study limitation which leads to greater variability (ie, unexplained error) in our modelling. This precludes us from making any conclusions around vaccination status and mortality, and likely prevented us from discovering additional associations between pre-existing conditions and COVID-19 mortality. However, given that the exposure of interest (pre-existing conditions) precedes the pandemic and the vaccines, vaccination status would not operate as a confounder of these associations—its exclusion would not differentially affect exposure status, and therefore not bias our estimates of effect. We have added to our discussion of this limitation and its likely impact on our findings (lines 343-356).

Point 2: Another important bias is related to the definition of the cause of death: the gold standard method is the autopsy tool (see DOI: 10.3390/medicina57040309). The authors missed describing if they consulted the autopsy report for each case classified as death from COVID. Please, clarify this important aspect. If an autopsy is not performed, discussed this bias in the discussion section.

Response 2: In the state of Colorado, autopsies are done on a case-by-case basis, but usually only occur in cases where the cause of death is uncertain. As described in the Methods (161-173), the information on cause of death for this study comes from the Colorado Department of Health and Environment (CDPHE) Vital Statistics Program which derives statistics from official records of vital events, which includes death certificate information. In order to identify all COVID-19-related deaths for analysis, we searched the death certificate data for both the ICD-10 code indicating death from COVID-19 as well as key words in the additional causes of death category (lines 166-170). It is possible COVID-19 attributable deaths were incorrectly or incompletely captured in the vital statistics records, as we now point out in the discussion (lines 325-329).

Point 3: Finally, the authors missed indicating how the patients were tested to determine the COVID-19 infection: each method could have a bias. If this data is not available, the authors should discuss this important point.

Response 3: The exact testing method is not included with COVID-19 test results in the patient information. The discussion was updated to include this information and its possible effect on the study (lines 321-322).

Point 4: Introduction section- Please, update the data reported in line 32 to the end of 2022.

Response 4: Lines 32-33 have been updated to reflect the most recent data available.

Point 5: Introduction section- Please, insert several considerations about the vaccination program: fortunately, it has influenced the mortality rate.

Response 5: As suggested, we consider the known influence of COVID-19 vaccines on mortality rates in the introduction section (lines 33-35, 78-81), and enhanced our discussion to better explain the impact of not having vaccination status in our study (see response to point 1 above).

Point 6: Material and methods- Please, insert the start and stop points in the collected data.

Response 6: As suggested, we added start and stop points for the collected data on line 172 in the methods section.

Point 7: Results- Table 1, Blood type: <10 -> it is incorrect. Please, insert the exact number.

Response 7: Cell counts with less than ten patients are censored to preserve patient privacy in accordance with CCPM Biobank policy. Therefore, the exact number of patients for blood types AB and B are not able to be reported and are instead designated as “<10”.

Point 8: Results- Blood type: several important data are missed to achieve a solid scientific conclusion. I suggest removing this section. Alternatively, it should be completely discussed this aspect.

Response 8: Although there was not enough data to come to a solid scientific conclusion, we believe it is still important to the overall literature surrounding COVID-19 mortality to include blood type as an important variable in addition to health record information. As suggested, we’ve more thoroughly discussed the importance of blood type and genetic information in the discussion (lines 382-387).

Point 9: Discussion- Please, improve this section. Particularly, I suggest inserting a specific section named "Limitations".

Response 9: In response to many reviewer suggestions, we made substantial changes that improve the content and organization of the discussion section (lines 325-329, 337-340, 344-356), including better discussion of limitations around vaccination status, measurement error in testing and deaths, and implications of these limitations on our conclusions.

Round 2

Reviewer 3 Report

Following the reviewers' suggestions, the authors have improved their manuscript.

Author Response

Point 1: Following the reviewers' suggestions, the authors have improved their manuscript.

Response 1: Thank you for your time and for the revisions you have provided to improve our manuscript.